# VALERIE: Visual-based inspection of alternative splicing events at single-cell resolution

**Wei Xiong Wen**[1,2], **Adam J. Mead**[1,3], **Supat Thongjuea**[2,3]*

**1** MRC WIMM Molecular Haematology Unit, MRC Weatherall Institute of Molecular Medicine, John Radcliffe Hospital, University of Oxford, Oxford, United Kingdom, **2** MRC WIMM Centre for Computational Biology, MRC Weatherall Institute of Molecular Medicine, John Radcliffe Hospital, University of Oxford, Oxford, United Kingdom, **3** NIHR Oxford Biomedical Research Centre, John Radcliffe Hospital, University of Oxford, Oxford, United Kingdom

* supat.thongjuea@imm.ox.ac.uk

**Data Availability Statement:** All relevant data are within the manuscript and its Supporting Information files. The software is available on the Comprehensive R Archive Network (CRAN):

## Abstract

We present VALERIE (Visualising alternative splicing events from single-cell ribonucleic acid-sequencing experiments), an R package for visualising alternative splicing events at single-cell resolution. To explore any given specified genomic region, corresponding to an alternative splicing event, VALERIE generates an ensemble of informative plots to visualise cell-to-cell heterogeneity of alternative splicing profiles across single cells and performs statistical tests to compare percent spliced-in (PSI) values across the user-defined groups of cells. Among the features available, VALERIE displays PSI values, in lieu of read coverage, which is more suitable for representing alternative splicing profiles for a large number of samples typically generated by single-cell RNA-sequencing experiments. VALERIE is available on the Comprehensive R Archive Network (CRAN): https://cran.r-project.org/web/packages/VALERIE/index.html.

This is a *PLOS Computational Biology* Software paper.

## Introduction

Technological advances in high-throughput next-generation transcriptomic sequencing have unravelled gene expression signatures underpinning physiological and pathological processes [1–3]. Alternative splicing represents an additional and underappreciated layer of complexity underlying gene expression profiles [4–6]. To date, alternative splicing has primarily been investigated using bulk RNA-sequencing. Alternative splicing analysis at single-cell resolution is an emerging area of research and it promises to provide novel biological insights previously missed by bulk RNA-sequencing [7, 8]. Notably, single-cell analysis showed differential isoform usage within apparently homogenous cell populations and revealed hidden subpopulation of cells that could not be distinguished by conventional approaches such as using cell-surface markers [8–11]. For example, single-cell analysis of mouse embryonic stem cells

https://cran.r-project.org/web/packages/VALERIE/index.html.

**Funding:** The Clarendon Fund and Oxford-Radcliffe Scholarship in conjunction with WIMM Prize PhD Studentship to W.X.W., Medical Research Council (MRC) Senior Clinical Fellowship and CRUK Senior Cancer Research Fellowship to A.J.M., and Oxford-Bristol Myers Squibb (BMS) Fellowship to S.T. The funders had no role in study design, data collection and analysis, decision to publish, or preparation of the manuscript.

**Competing interests:** The authors have declared that no competing interests exist.

(ESCs) revealed differential variation in isoform usage reflecting changes in the dynamic state of cells such as the cell cycle [11].

To explore alternative splicing events, the visual-based inspection of sequencing read coverage in the genome browser is the usual practice prior or complementary to laboratory-based validation of gene or alternative splicing profiles, such as using quantitative polymerase chain reaction (RT-qPCR) or single-molecule fluorescence *in situ* hybridisation (smFISH). Existing genome browser visualisation tools are optimized for small-scale bulk RNA-sequencing datasets [12–14]. Present approaches for visualising alternative splicing events in single cells include aggregating single cells by cell types [14, 15], presenting only a subset of single cells [16], or presenting all single cells in the study [8, 17]. The first and second approaches do not capture cell-to-cell heterogeneity in the entire population of cells, which is a key component of single-cell data, whereas the third approach is difficult to delineate overall alternative splicing patterns across different cell populations. Moreover, read coverage which is commonly employed to visualise gene expression profiles is not suitable for representing alternative splicing events. Read coverage distribution across a genomic locus is represented as bar graphs where the height of the bar graphs is proportional to the number of sequencing reads spanning across the genomic locus [13].

Percent spliced-in (PSI) is defined as the fraction of sequencing reads supporting the included isoform or exon and is more relevant for representing alternative splicing profiles [18, 19]. For example, a PSI of 0.50 (or 50%) means half of all sequencing reads support the included isoform or exon. In addition to read coverage, sashimi plots include arcs connecting two splice sites and the width of the arcs is proportional to the number of sequencing reads supporting the splice junctions [12–14]. Nevertheless, the percentage, instead of the absolute number of sequencing reads, connecting two splice sites may be more intuitive to interpret the degree of exon inclusion.

An ideal tool for visualising single-cell alternative splicing data would incorporate a quantification of PSI across all individual single cells (not a single cell ensemble) with a statistical test to identify significant differences between cell populations. Here, we introduce VALERIE, a tool which incorporates these features, to complement existing next-generation sequencing visualisation platforms for inspecting cell-to-cell heterogeneity of alternative splicing events across different cell populations at single-cell resolution.

## Design and implementation

In DNA-sequencing, split reads indicate genetic mutations, specifically deletions. In RNA-sequencing, split reads spanning a genomic locus indicate alternative splicing events such as exon-skipping. Therefore, split reads, in addition to read coverage, need to be taken into consideration for alternative splicing analysis and consequently visualisation of the events [20–22]. VALERIE is implemented in R. It requires sorted binary alignment map (BAM) files with their corresponding index files, sample information file specifying BAM file names and user-defined cell groups for each single cell, and exon information file specifying the types of alternative splicing events and their corresponding genomic coordinates as the input. Single-cell groups can be known *a priori* using fluorescence-activated cell sorting (FACS) gating of cell-surface markers or can be assigned using unsupervised clustering approaches and gene signatures from RNA-sequencing data [9, 10]. Genomic coordinates corresponding to alternative splicing events can be extracted from the splicing detection and quantitation tools such as BRIE and MISO [12, 16]. VALERIE supports different types of alternative splicing events. These include skipped-exon (SE), mutually exclusive exons (MXE), retained-intron (RI), alternative 5' splice site (A5SS), and alternative 3' splice site (A3SS).

Implementations that interpret gapped alignments containing split reads are required because split reads are needed to infer degree of exon inclusion. In SAM/BAM format, split reads are characterised by reference skipping cigar operations ("N"). Hence, implementations which are able to identify these features are required to infer alternative splicing events. To this end, we implemented the function 'readGAlignments' [23] to take gapped alignments into account when reading BAM files. To enable fast and efficient reading of BAM files, we implemented the function 'ScanBamParam' based on the GenomicRanges [23] to selectively read regions corresponding to the input information of alternative splicing events as defined in the user-supplied exon information file. For each single cell, PSI values at each genomic coordinate supported by at least 10 reads are then computed as the proportion of non-split reads over the total number of reads, where the total number of reads is the sum of non-split reads and split reads [24, 25]. PSI values at each genomic coordinate across alternative exon and its constitutive exons for all single cells are represented in a heatmap plot. A line graph summarises the PSI values across all single cells at each genomic coordinate across user-defined groups of single cells by using the mean as the summary statistic. Significant differences of PSI values at each genomic coordinate between user-defined groups of single cells can be assessed using the t-test and Wilcoxon rank-sum test for two-group comparison. Analysis of variance (ANOVA) and Kruskal-Wallis test can be performed for more than two-group comparison. P-values can be corrected for multiple testing using the 'p.adjust' function options. Mean PSI and p-value line graphs are implemented using ggplot2 package. The workflow of data processing steps is described in S1A Fig. For each alternative splicing event, VALERIE generates an image consisting of three graphical components as shown in Fig 1, i.e. heatmap of PSI values from each single cell classified by a user-defined group of single cells, and mean PSI and p-value line graphs along genomic coordinates per event for groups of single cells.

VALERIE enables visual inspection of alternative splicing events identified from the genome-wide differential analysis of different groups of single cells. Candidate events can be visualised simultaneously. VALERIE may serve as a complementary, or an alternative, approach to validate alternative splicing events identified from single-cell short-read RNA-sequencing such as RT-qPCR and smFISH. Verified alternatively splicing events can subsequently be prioritised for downstream functional studies (S1B Fig).

## Features

VALERIE provides several unique features to complement existing visualisation platforms for inspecting alternative splicing events at single-cell resolution:

a. Displays PSI values for single cells instead of read coverage. PSI value is more suitable for representing alternative splicing profiles whereas read coverage is more suitable for representing gene expression profiles. It is widely reported that changes in alternative splicing profiles (differential isoform usage) were not always accompanied by changes in gene expression profiles [11, 26, 27]. For example, differential isoform usage analysis revealed differential expression of *CD45* between CD45RO[+] memory and CD45RA[+] naïve T cell populations. However, analysis using gene counts did not detect differential *CD45* usage between these two cell populations [27].

b. Displays PSI values at each genomic coordinate along the alternate exon and flanking constitutive exons. This is in contrast to current approaches that limit splice information to exon-exon junctions [13, 14]. Presenting PSI values at each genomic coordinate enables consistency of PSI profile across the entire exon length to be assessed. This is relevant for sequencing approaches that utilise very short reads, e.g. 50bp single-end reads that do not span the entire exon length (S2 Fig) [9].

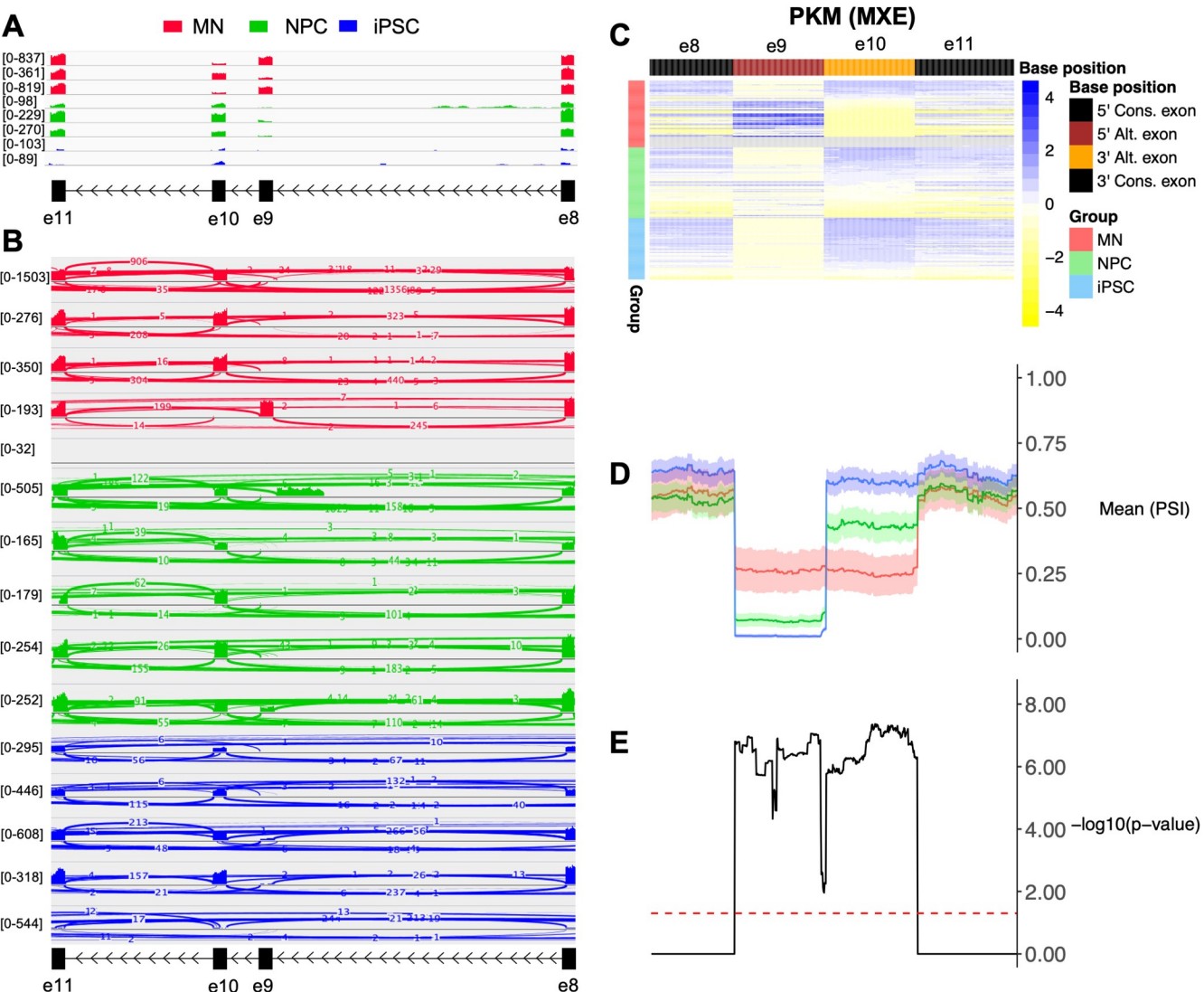

**Fig 1. Read coverage and percent spliced-in (PSI) profile of *PKM* mutually exclusive exons 9 and 10 in single cells from 63 induced pluripotent stem cells (iPSCs), 69 motor neuron cells (MNs), and 73 neural progenitor cells (NPC)** [8]. **(A)** Integrative Genome Browser (IGV) presentation of read coverage from bulk RNA-sequencing data. MNs showed different read coverage at exons 9 and 10 across replicates whereas NPCs and iPSCs showed consistently higher read coverage at exon 10 compared to exon 9 across all replicates. **(B)** IGV sashimi plot of selected single-cell RNA-sequencing data. Heterogeneity observed in relative read coverage at exons 9 and 10 across single cells with some single cells with low-to-no coverage. **(C-E)** VALERIE presentation of PSI values from entire single-cell RNA-sequencing data. **(C)** Heatmap unravelled 2 subpopulations of MNs and NPCs whereby one population exclusively expressed exon 9 while another subpopulation exclusively expressed exon 10, whereas iPSCs consist of a single homogenous population that exclusively expressed exon 10. **(D)** Mean PSI values across the genomic coordinates corresponding to the flanking constitutive exons and mutually exclusive exons. Overall, iPSCs showed decreased usage of exon 10 and increased usage of exon 9 after differentiating into MNs or NPCs. MNs showed similar usage of exons 9 and 10 whereas NPCs showed higher usage of exon 10 compared to exon 9. Shaded regions represent 95% confidence interval (CI) of the mean. **(E)** Differences in mean PSI values across iPSCs, MNs, and NPCs, were statistically significant at the genomic coordinates corresponding to alternative splicing event (mutually exclusive exons) but were, as expected, not statistically significant at the genomic coordinates corresponding to the flanking constitutive exons. P-values were computed using Kruskal-Wallis test and adjusted for multiple testing using Bonferroni correction. The red dashed line indicates -log10 of the p-value of 0.05. Colour bar indicates scaled PSI values (z-scores) across rows (single cells). Grey regions in the heatmap indicate genomic positions with less than 10x coverage. Alt. exon: Alternatively spliced exon. Cons. exon: Constitutive exon. MXE: Mutually exclusive exons.

 c. Summarises PSI profiles for user-defined groups of single cells.

 d. Performs statistical tests to assess significant differences in PSI profiles between user-defined groups of single cells.

e.  Omits non-informative intronic regions, except in cases involving intron-retention alternative splicing events.

f.  Annotates and standardises relative positions of alternative and constitutive exons in 5'-to-3' transcription direction.

## Results

To enable validation of VALERIE, we identified high-quality single-cell alternative splicing events that have been validated using RT-qPCR and smFISH. We included single-cell RNA-sequencing dataset from 63 induced pluripotent stem cells (iPSCs), 69 motor neuron cells (MNs), and 73 neural progenitor cells (NPCs) from a previous study [8]. In this study, bulk RNA-sequencing data for each cell type were also available. Thus, we were able to compare visualisation of alternative splicing events at both single-cell and bulk levels using VALERIE and existing visualisation platforms. The pyruvate kinase M1/2 (*PKM*) gene produces two main transcripts *PKM1* and *PKM2* which are differentially expressed when iPSCs differentiate into MNs or NPCs. Specifically, iPSCs exclusively express exon 10 whereas MNs and NPCs consist of two subpopulations of cells that either express exon 9 or 10. Hence, the type of alternative splicing event *PKM* undergoes is mutually exclusive exons (MXE). Read coverage presentation of bulk RNA-sequencing data demonstrated exclusive exon 10 expression in iPSCs but simultaneous exon 9 and 10 expression in MNs and NPCs (Fig 1A). Sashimi plots of selected single cells unravelled cell-to-cell heterogeneity of exon usage between the different cell types but were unable to capture differential exon usage across all cells (Fig 1B). Moreover, comparison of separate plots for all cells may become overwhelming and ultimately aggregating single cells by cell types becomes necessary, but this will obscure cell-to-cell heterogeneity [14]. VALERIE captured cell-to-cell heterogeneity across all single cells and also enabled comparison of overall alternative splicing profiles across the different cell populations (Fig 1C–1E).

We used the STAR splice-aware aligner to map sequencing reads to the human reference genome [28]. STAR is the popular choice among published single-cell alternative splicing studies [8, 9, 15, 24]. HISAT2 (hierarchical indexing for spliced alignment of transcripts) is a splice-aware aligner with superior alignment speed compared to other aligners [29]. To evaluate the PSI profile generated by different aligners, we performed analysis on the same dataset using HISAT2. *PKM* mutually exclusive exons 9 and 10 similarly showed differential splicing across iPSCs, MNs, and NPCs (S3 Fig), albeit with a slightly higher level of mean PSI values and higher level of statistical significance (lower p-values) between cell types compared to the alignment using STAR. This is may be in part attributed to the differences in sensitivity and specificity between the two aligners [29]. We further subsampled aligned reads for each single cell to 50%, 25%, and 1% of the original read depth in order to simulate PSI profiles at varying read depths and gene expression levels. As expected, we observed higher power to detect differential *PKM* mutually exclusive exons 9 and 10 usage at higher read depth as reflected by the clearer signal of mean PSI and higher significant level at genomic coordinates corresponding to the alternatively spliced exons (S4 Fig). We investigated the PSI profiles for alternative splicing events at different positions, 3'-end, 5'-end, and in the middle of the transcript. We identified a representative alternative splicing event from each category from the same study that was validated using single-cell qPCR [8]. All alternatively spliced exons were observed to be differentially spliced across iPSCs, MNs, and NPCs (S5 Fig). Notably, constitutive exons which were either the last or first exon of the transcript showed varying coverage at the 3'-end and 5'-end, respectively (S5A and S5B Fig). Similar observations were reported previously from end-to-end sequencing of entire isoforms in single cells using long-read RNA-sequencing such as

Nanopore and PacBio [30–32]. Interestingly, VALERIE revealed an alternative 3' splice site (A3SS) on the 5' constitutive exon (exon 5) based on changes in PSI values (S5C Fig). Though this A3SS usage was not significantly different across the different cell types.

We measured the computational time to process single cells from investigating alternative splicing dynamics in mouse oligodendrocytes [9]. VALERIE required an average of 27.3s to process 2,000 single cells based on 10 repeated evaluations using the microbenchmark package. This evaluation was performed on an iMac with 3.5 GHz Quad-Core Intel Core i5 processer and 32 GB memory.

## Availability and future directions

Recent advances in full-length library preparation protocols enabled amplification and subsequently sequencing of small amount of starting RNA materials such as that from single cells [33, 34]. Current visualisation platforms are optimised for visualising gene and alternative splicing profiles for small-scale bulk RNA-sequencing datasets [12–14]. VALERIE complements existing implementations by enabling visualisation of alternative splicing events for single-cell RNA-sequencing datasets typically generated by full-length library preparation methods such as Smart-seq2 [34] and it is not appropriate for high-throughput droplet-based platforms such as the Chromium 10x genomics. It would be of particular interest to extend VALERIE's functionality to include visualisation of alternative splicing events from single-cell long-read RNA-sequencing datasets. VALERIE is available on the Comprehensive R Archive Network CRAN (https://cran.r-project.org/web/packages/VALERIE/index.html).

## Supporting information

**S1 Fig. Overview of VALERIE. (A)** The workflow of data processing steps. VALERIE computes percent spliced-in (PSI) values from read coverage information and integrates alternative splicing coordinates and cell group annotations from exon and sample information files to generate heatmap of PSI, and line graphs of mean PSI and adjusted p-values at each nucleotide position. **(B)** The role of VALERIE in the overall process of identifying candidate alternative splicing events for downstream functional studies. VALERIE serves as a visual inspection and validation of alternative splicing events identified from genome-wide analysis such as differential analysis. In conjunction with, or as an alternative to, other technical validation approaches as such sc-qPCR, VALERIE can enable selection of candidate alternative splicing events for downstream functional validation. A3SS: Alternative 3' splice site. A5SS: Alternative 5' splice site. MXE: Mutually exclusive exons. RI: Retained-intron. sc-qPCR: Single-cell quantitative polymerase chain reaction. SE: Skipped-exon. smFISH: Single-molecule fluorescence *in situ* hybridisation.
(TIFF)

**S2 Fig. Percent spliced-in (PSI) profile of *Mbp* exon 2 skipping in 46 single cells from mice induced with experimental autoimmune encephalomyelitis (EAE) and 57 single cells from healthy mice [9].** Top: Sparse profile of PSI values due to long exon lengths relative to sequencing reads. Here, exon 1, 2, and 3 are 171, 78, and 102 base-pair (bp) in length whereas libraries were sequenced in 50bp single-end mode. Middle: Mean PSI values across the genomic coordinates corresponding to the flanking constitutive exons and skipped-exon. Overall, single cells from EAE mice showed increased exon 2 usage compared to single cells from control mice. Bottom: Differences in mean PSI values across EAE and control cell groups were statistically significant at the genomic coordinates corresponding to alternative splicing event (skipped-exon) but were, as expected, not statistically significant at the genomic coordinates

corresponding to the flanking constitutive exons. P-values were computed using Wilcoxon rank-sum test and adjusted for multiple testing using Bonferroni correction. The red dashed line indicates -log10 of the p-value of 0.05. Colour bar indicates scaled PSI values (z-scores) across rows (single cells). Grey regions in the heatmap indicate genomic positions with less than 10x coverage. Alt. exon: Alternatively spliced exon. Cons. exon: Constitutive exon. SE: Skipped-exon.
(TIFF)

**S3 Fig. Percent spliced-in (PSI) profile of *PKM* mutually exclusive exons 9 and 10 in single cells from 63 induced pluripotent stem cells (iPSCs), 69 motor neuron cells (MNs), and 73 neural progenitor cells (NPC).** Sequencing reads were aligned using HISAT2 [29] in lieu of STAR [28] as in Fig 1. Alignment with HISAT2 similarly showed significant differential *PKM* mutually exclusive exon usage across the three cell populations. P-values were computed using Kruskal-Wallis test and adjusted for multiple testing using Bonferroni correction. The red dashed line indicates -log10 of the p-value of 0.05. Colour bar indicates scaled PSI values (z-scores) across rows (single cells). Grey regions in the heatmap indicate genomic positions with less than 10x coverage. Alt. exon: Alternatively spliced exon. Cons. exon: Constitutive exon. MXE: Mutually exclusive exons.
(TIFF)

**S4 Fig. Percent spliced-in (PSI) profile of *PKM* mutually exclusive exons 9 and 10 in single cells from 63 induced pluripotent stem cells (iPSCs), 69 motor neuron cells (MNs), and 73 neural progenitor cells (NPC) at different read depth.** Aligned sequencing reads were sub-sampled to yield read depth of **(A)** 50%, **(B)** 25%, and **(C)** 1% of the original read depth to simulate PSI profile at different read depth. P-values were computed using Kruskal-Wallis test and adjusted for multiple testing using false discovery rate (FDR). The red dashed line indicates -log10 of the p-value of 0.05. Colour bar indicates scaled PSI values (z-scores) across rows (single cells). Grey regions in the heatmap indicate genomic positions with less than 10x coverage. Alt. exon: Alternatively spliced exon. Cons. exon: Constitutive exon. MXE: Mutually exclusive exons.
(TIFF)

**S5 Fig. Percent spliced-in (PSI) profiles of alternatively spliced exons located at 3'-end, 5'-end, and in the middle of the transcript. (A)** *RPS24* alternatively spliced exon positioned at the 3'-end (2nd last exon) of the ENST00000435275.5_4 transcript consisting 6 exons in length. **(B)** *RBPJ* alternatively spliced exon positioned at the 5'-end (2nd exon) of the ENST00000355476.7_2 transcript consisting 12 exons in length. **(C)** *DYNC1I2* alternatively spliced exon positioned in the middle (6th exon) of the ENST00000355476.7_2 transcript consisting 18 exons in length. Differences in PSI values on the *DYNC1I2* 5' constitutive exon (exon 5) revealed an alternative 3' splice site (A3SS) located on the exon. This A3SS is annotated in GENCODE. Transcript IDs correspond to GENCODE v34lift37. P-values were computed using Kruskal-Wallis test and adjusted for multiple testing using Bonferroni correction. The red dashed line indicates -log10 of the p-value of 0.05. Colour bar indicates scaled PSI values (z-scores) across rows (single cells). Grey regions in the heatmap indicate genomic positions with less than 10x coverage. Alt. exon: Alternatively spliced exon. Cons. exon: Constitutive exon. SE: Skipped-exon.
(TIFF)

**S1 File. VALERIE source code, documentation, and test data**
(GZ)

## Acknowledgments

The authors would like to thank the members of WIMM Centre for Computational Biology for testing the VALERIE package and providing useful input.

## Author Contributions

**Conceptualization:** Wei Xiong Wen, Adam J. Mead, Supat Thongjuea.

**Software:** Wei Xiong Wen.

**Supervision:** Adam J. Mead, Supat Thongjuea.

**Writing – original draft:** Wei Xiong Wen.

**Writing – review & editing:** Adam J. Mead, Supat Thongjuea.

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
