## [Decision Letter · Decision Letter 0]

23 May 2020

Dear Dr. thongjuea,

Thank you very much for submitting your manuscript "VALERIE: Visual-based inspection of alternative splicing events at single-cell resolution" for consideration at PLOS Computational Biology. As with all papers reviewed by the journal, your manuscript was reviewed by members of the editorial board and by several independent reviewers. The reviewers appreciated the attention to an important topic. Based on the reviews, we are likely to accept this manuscript for publication, providing that you modify the manuscript according to the review recommendations.

Sincerely,

Mihaela Pertea

Software Editor

PLOS Computational Biology

Mihaela Pertea

Software Editor

PLOS Computational Biology

[LINK]

Reviewer's Responses to Questions

**Comments to the Authors:**

Reviewer #1: In this manuscript, Wen and colleagues describe a new tool for the visualization of alternative splicing events at single-cell resolution. They have named it VALERIE and they have made it available as an R package on CRAN. The authors have provided an example of the ability of VALERIE (compared to other platforms) to visualize an alternative splicing event at both single-cell and bulk levels using RNA sequencing data from the study by Song et al (Mol Cell 2017) on induced pluripotent stem cells, motor neuron cells and neural progenitor cells. Overall, VALERIE has several useful features that complement existing visualization platforms and is a valuable addition to the relatively small toolbox currently available for the visualization of alternative splicing events from RNA-seq data.

Comments

- The authors have demonstrated the ability of VALERIE to visualize alternative splicing events and perform statistical tests using a single example of mutually exclusive exons (MXE) in the PKM gene in iPSCs and neuron cells. It would be interesting to see the visualization of at least another alternative splicing event (preferably a different type from MXE) either from the same study or from a different suitable study. Another example is shown in Figure S1, but this was specifically presented for data including very short reads.

- Line 82: The authors should specify which panel of Figure 1 refers to the image generated by VALERIE.

- In Figure 1 and Figure S1, it would be helpful to add a short text describing what the heatmap shows (i.e. PSI values) next to the colorbar.

- In the Introduction, the syntax in line 10 should be corrected.

Reviewer #2: The article by Wei Xiong Wen describes a novel tool for visualisation of alternative splicing events in single-cell RNA sequencing data.

Alternative splicing is an important mechanism for the generation of proteomic diversity from the genome, and is critically overlooked in most single-cell RNA-sequencing studies. Tools that assist in the detection and visualisation of alternative splicing events from single-cell data are therefore, very welcome.

The VALERIE R package enables visualisation of percent spliced in (PSI) in lieu of read coverage to enable visualisation of a number of single-cell samples in parallel, using BAM files as an input.

The paper is concise, and the tool itself is useful – I have a number of relatively minor comments I would like to see addressed in a final publication.

1) How can this tool be used in a discovery process? The authors show that it can potentially resolve alternative splicing events in individual genes, but it would be important that the authors demonstrate how the tool can be used in conjunction with others to identify alternative splicing events from a large dataset and rank them on the basis of supporting evidence. A graphical overview of the data processing would be helpful.

2) It must be made clearer that this method appropriate primarily for plate-based methods such as Smart-seq2 which capture full-length sequence (although this is fragmented for sequencing) and not appropriate for e.g. 10x genomics libraries.

3) Is there a hypothetical maximum number of cells the tool can cope with?

4) Are there any mapping constraints for the method – i.e. are particular aligners more favourable than others. Is there any other pre-processing that might affect the outcome of the analysis?

5) Similarly it would be essential to reflect on any coverage constraints for detection – e.g. how does the tool fare with genes at different expression levels (perhaps percentiles of overall expression levels) or indeed genes for which splicing events are located at the 3’, mid or 5’ of the transcript. I understand that some limitations here will be due to the biology/molecular biology preceding this analysis but it is important for potential users of the tool to understand potential limitations when thinking about this kind of analysis.

6) The main figure is quite basic and cluttered in appearance, panels C should be split into C, D and E and some additional explanation in the figure legend.

**Have all data underlying the figures and results presented in the manuscript been provided?**

Reviewer #1: Yes

Reviewer #2: Yes

PLOS authors have the option to publish the peer review history of their article (what does this mean?). If published, this will include your full peer review and any attached files.

Reviewer #1: No

Reviewer #2: No
---

## [Decision Letter · Decision Letter 1]

26 Jul 2020

Dear Dr. thongjuea,

We are pleased to inform you that your manuscript 'VALERIE: Visual-based inspection of alternative splicing events at single-cell resolution' has been provisionally accepted for publication in PLOS Computational Biology.

Best regards,

Mihaela Pertea

Software Editor

PLOS Computational Biology

Mihaela Pertea

Software Editor

PLOS Computational Biology

Reviewer's Responses to Questions

**Comments to the Authors:**

Reviewer #1: The authors have addressed all my comments. I have no further comments.

**Have all data underlying the figures and results presented in the manuscript been provided?**

Reviewer #1: Yes

PLOS authors have the option to publish the peer review history of their article (what does this mean?). If published, this will include your full peer review and any attached files.

Reviewer #1: No

---

## [Editor Report · Acceptance letter]

28 Aug 2020

PCOMPBIOL-D-20-00566R1 

VALERIE: Visual-based inspection of alternative splicing events at single-cell resolution

Dear Dr thongjuea,

I am pleased to inform you that your manuscript has been formally accepted for publication in PLOS Computational Biology. Your manuscript is now with our production department and you will be notified of the publication date in due course.

With kind regards,

Laura Mallard
